# An Open One-Step RT-qPCR for SARS-CoV-2 detection

Ariel Cerda[1,2⦿], Maira Rivera[1,3⦿], Grace Armijo[1,2], Catalina Ibarra-Henriquez[1,2], Javiera Reyes[1,3], Paula Blázquez-Sánchez[1,3], Javiera Avilés[1], Aníbal Arce[1], Aldo Seguel[1], Alexander J. Brown[4,5], Yesseny Vásquez[6], Marcelo Cortez-San Martín[7], Francisco A. Cubillos[1,7], Patricia García[8], Marcela Ferres[8], César A. Ramírez-Sarmiento[1,3]*, Fernán Federici[1,2,3]*, Rodrigo A. Gutiérrez[1,2]*

1 ANID—Millennium Science Initiative Program—Millennium Institute for Integrative Biology (iBio), Santiago, Chile, 2 FONDAP Center for Genome Regulation, Departamento de Genética Molecular y Microbiología, Pontificia Universidad Católica de Chile, Santiago, Chile, 3 Institute for Biological and Medical Engineering, Schools of Engineering, Medicine and Biological Sciences, Pontificia Universidad Católica de Chile, Santiago, Chile, 4 Department of Biomedical Research, National Jewish Health, Denver, CO, United States of America, 5 Department of Immunology & Microbiology, University of Colorado Anschutz Medical Campus, Aurora, CO, United States of America, 6 Escuela de Ciencias Médicas, Facultad de Medicina, Universidad de Santiago de Chile, USACH, Santiago, Chile, 7 Departamento de Biología, Facultad de Química y Biología, Universidad de Santiago de Chile, USACH, Santiago, Chile, 8 Departamento de Laboratorios Clínicos, Escuela de Medicina, Facultad de Medicina, Pontificia Universidad Católica de Chile, Santiago, Chile

⦿ These authors contributed equally to this work.
* cesar.ramirez@uc.cl (CARS); ffederici@bio.puc.cl (FF); rgutierrez@bio.puc.cl (RAG)

**Data Availability Statement:** We confirm that all data generated during this study, including both edited and unedited images, are included within the published article and its Supplementary Information files and have been deposited in the

## Abstract

The COVID-19 pandemic has resulted in millions of deaths globally, and while several diagnostic systems were proposed, real-time reverse transcription polymerase chain reaction (RT-PCR) remains the gold standard. However, diagnostic reagents, including enzymes used in RT-PCR, are subject to centralized production models and intellectual property restrictions, which present a challenge for less developed countries. With the aim of generating a standardized One-Step open RT-qPCR protocol to detect SARS-CoV-2 RNA in clinical samples, we purified and tested recombinant enzymes and a non-proprietary buffer. The protocol utilized M-MLV RT and Taq DNA pol enzymes to perform a Taqman probe-based assay. Synthetic RNA samples were used to validate the One-Step RT-qPCR components, demonstrating sensitivity comparable to a commercial kit routinely employed in clinical settings for patient diagnosis. Further evaluation on 40 clinical samples (20 positive and 20 negative) confirmed its comparable diagnostic accuracy. This study represents a proof of concept for an open approach to developing diagnostic kits for viral infections and diseases, which could provide a cost-effective and accessible solution for less developed countries.

## Introduction

The severe acute respiratory syndrome coronavirus 2 (SARS-CoV-2), the etiological agent of the coronavirus disease 2019 (COVID-19) [1], has infected more than 770 million individuals and killed over 6.9 million people worldwide [2]. However, the actual number of cases is likely

Zenodo open data repository [https://doi.org/10.5281/zenodo.7551378]. Furthermore, to enhance the reproducibility of our study, we are also providing access to the three protein purification protocols standardized and utilized in our research through the following links on protocols.io. Pfu-Sso7d: https://dx.doi.org/10.17504/protocols.io.bzusp6we M-MLV RT: dx.doi.org/10.17504/protocols.io.bsernbd6 Taq DNA pol: dx.doi.org/10.17504/protocols.io.bya3psgn.

**Funding:** This work was supported by the National Agency for Research and Development (ANID) through the ANID Millennium Science Initiative Program (ICN17_022) to RAG, CARS and FF. Fondo de Desarrollo de Areas Prioritarias (Center for Genome Regulation; ANID/FONDAP/15090007), to RAG and FF. Fondo de Desarrollo Científico y Tecnológico (FONDECYT 1201684 awarded to CARS, FONDECYT Regular 1211218 to FF, and FONDECYT 3190731 to MR), and an International Cooperation Program with Consejo Nacional de Ciencia, Tecnología e Innovación Tecnológica (ANID-CONCYTEC covbio0012 awarded to FF and CARS). This work was also supported by the National Institutes of Health NIAID training grant (Training Program in Immunology; T32-AI07405) awarded to AJB. PBS and JR were supported by ANID Doctoral Scholarships (21191979 and 21191684, respectively). The funders had no role in study design, data collection and analysis, decision to publish, or preparation of the manuscript.

**Competing interests:** The authors have declared that no competing interests exist.

higher, with recent estimates from the WHO suggesting that excess mortality caused by the pandemic may be as high as 14.9 million [3]. Oversaturation of local health systems and persistent scarcity of testing supplies and personal protective equipment (PPE), particularly in developing countries, have hindered early diagnosis and impeded the control and monitoring of virus transmission [4–7]. This has resulted in significant underestimation of the number of infected individuals, posing challenges for the efficient management of health resources [8, 9].

To confirm suspected cases of COVID-19, various diagnostic systems have been proposed, including molecular, serological, viral culture, and chest computed tomography imaging methods [10–14]. While each of these methods provides valuable information to guide public healthcare decision-making, real-time reverse transcription polymerase chain reaction (RT-PCR) remains the gold standard approach for SARS-CoV-2 diagnosis due to its high sensitivity, specificity, and rapid detection [15–19]. One-Step RT-qPCR is the most commonly used technique for SARS-CoV-2 diagnosis and is recommended by health authorities worldwide [20, 21]. The CDC-approved One-Step RT-qPCR Diagnostic Panel employs oligonucleotides that have been tested to avoid cross-reactivity with other human coronaviruses or common respiratory pathogens. Additionally, the performance of these oligonucleotides is continually monitored to ensure their efficacy against new variants of concern (VOC), such as the Delta and Omicron variants [21–25].

The unprecedented amount of testing needed during the COVID-19 pandemic, particularly during the first few months, caused supply-chain shortages that prevented researchers and clinicians worldwide from performing testing by RT-qPCR. As a result, numerous initiatives were undertaken to overcome this problem [26–29]. Most approaches focused on the two steps that were most affected by the availability of diagnostic testing supplies: viral RNA extraction from patient samples and amplification/detection systems. Simplifying RNA extraction methods addressed the shortage of RNA extraction kits by pooling nasopharyngeal samples [30–34] or using alternative homemade simple RNA extraction methods [35–38]. Some reports described direct testing protocols without RNA extraction or replacing the extraction with simple chemical or temperature treatment of the samples [27, 39–43]. Unfortunately, finding alternatives for the reagents used in the detection and amplification of viral RNA is difficult, mainly because they have a higher cost, require more specialized infrastructure, and the enzymes used in diagnostic protocols are usually subject to centralized production models and intellectual property restrictions [44, 45]. This issue has become particularly relevant for low-middle income countries (LMIC), particularly in the Southern Hemisphere, where reagents are imported, and local means to produce them are scarce [46, 47]. Thus, strategies are required to ensure the access, production, and transfer of these components between researchers to facilitate the tracing and testing of emerging pathogens in LMICs.

Open-source communities have been promoting global collaboration, more accessible solutions and distributed responses since the beginning of the COVID-19 pandemic. They have offered low-cost alternatives to commercial reagents, free and open-source scientific and medical Hardware (FOSH) and PPE [36, 48–51]. In the context of diagnostic supplies, the goal is to replace expensive commercial kits with master mixes made up of easy-to-obtain reagents and homemade enzymes that can be produced and implemented locally and at low cost. Currently, few open protocols or resources are available that grant autonomy or cost reduction to LMIC in COVID-19 diagnostic methods [36, 52–55]. For instance, Graham et al. (2021) generated straightforward protocols for RNA extraction and RT-qPCR, which provides an open-source master mix for detecting SARS-CoV-2 with results comparable to commercial kits in nasopharyngeal swab samples and a detection limit of ~50 RNA copies. Furthermore, a "blueprint" to replicate RT-qPCR kits in academic laboratories has been proposed to overcome diagnostic supply shortages. This can be modified and adapted to resource-limited settings [54]. Finally,

the use of enzymes with reverse transcriptase (RT) and DNA polymerase activity (DNA pol) for one-enzyme RT-qPCR approaches has also been explored. For example, there is a dye-based RT-qPCR assay that employs the thermostable reverse transcriptase/DNA polymerase (RTX) [52, 54, 56] and a probe-based RT-qPCR assay that exploits the intrinsic RT activity of Taq DNA polymerase [57]. Despite these efforts, more RT-qPCR approaches to detect SARS-CoV-2 using non-commercial, readily procurable reagents, and easily shareable off-patent enzymes are still needed.

With the aim of contributing to these efforts, we present a standardized One-Step open RT-qPCR protocol that is based on the local production of recombinant enzymes available in the public domain, along with the generation of a non-proprietary buffer. The protocol outlined here allows for performing a Taqman probe-based assay using the Moloney Murine Leukemia Virus reverse transcriptase (M-MLV RT) [55] and Taq DNA polymerase (Taq DNA pol) enzymes [36]. The open RT-qPCR master mix described herein was evaluated on synthetic RNA and synthetic samples and did not show significant differences compared to the commercial RT-qPCR kit. Furthermore, detailed protocols for the purification of the RT and DNA pol enzymes, as well as the preparation of the homemade reaction buffer, are provided. This work, along with the genetic resources for the production and subsequent purification of the enzymes used in this study, is part of the Reagent Collaboration Network or ReClone (https://reclone.org), which ensures open and expeditious access to resources through an open material transfer agreement (openMTA) [58].

## Materials and methods

### Biological samples

Clinical samples analyzed in this study were primarily sourced from two distinct entities and were pre-evaluated using commercial kits to provide a benchmark for our custom master mix assays.

Forty nasopharyngeal swabs were collected from potential COVID-19 patients in the Metropolitan Region of Chile, from 28 December 2020 to 9 April 2021, as part of an active surveillance effort and processed in the Molecular Virology Laboratory at the University of Santiago of Chile. RNA was extracted using the E.Z.N.A. Total RNA Kit I (Omega Biotek, catalogue number R6834-02) according to the manufacturer's instructions. The SARS-CoV-2 virus was detected using the commercial GenomeCoV19 Detection Kit (Applied Biological Materials Inc., catalogue number G628.v2) in an Aria Mx Thermocycler (Agilent) with the thermal cycle program suggested by Applied Biological Materials (ABM). Samples with amplifications in both the N and S genes were considered positive according to the manufacturer's instructions. All methods and procedures were performed in accordance with ethical guidelines, under the authorization of the Ministry of Health and Universidad de Santiago de Chile (Health Alert Decree N˚4 of 2020), safeguarding the rights and well-being of the participants involved. The samples involved in our research underwent anonymization by the providing entity before any analysis, upholding the highest standards of privacy and confidentiality. On the other hand, 33 nasopharyngeal swabs in Universal Transport Medium (UTM) were collected from 17 to 22 July 2020 at the UC-Christus Health Network in Chile, from individuals who presented with symptoms suggestive of COVID-19 and were seeking medical consultation. RNA was automatically extracted using the Mag-Bind RNA Extraction Kit (Maccura Biotechnology CO., LTD, catalogue number GN7102903) according to the manufacturer's instructions. The samples were tested using the RT-qPCR commercial kit LightMix® SARS-CoV-2 RdRp-gene EAV PSR & Ctrl (TIB MOLBIOL, catalogue number 53-0777-96) in a LightCycler® 480 real-time PCR system (Roche). The data was analyzed using the 2nd Derivative Maximum Method

to obtain the quantification cycle (Cq) value for each sample. The samples were processed in the Clinical Laboratory of Medicine School. Of these, 24 were later confirmed as COVID-19 positive, while the remainder tested negative. The Ethical Review Board of the Faculty of Medicine, Pontificia Universidad Católica de Chile (code: 210105007) approved the use of these biological samples for diagnostic tests. An amendment to the original resolution allows for the use of the samples without directly obtaining informed consent, given that they have been anonymized by the providing entity to ensure the utmost privacy and confidentiality of the participants.

## Enzyme cloning, expression and purification

The plasmid encoding the codon-optimized Pfu-Sso7d was obtained from Dr. Alexander Klenov at York University. The E602D mutant Taq polymerase plasmid was obtained from Dr. Robert Tjian and his sequence is available in Addgene (Addgene plasmid # 166944; http://n2t. net/addgene:166944). The M-MLV RT-encoding plasmid was synthesized as a gBlock (IDT) with an additional N-terminal region of 8 amino acids containing a peptide leader. This M-MLV contains point mutations D200N, L603W, T330P, L139P, and E607K shown to increase thermostability and processivity [59]. The enzymes *Nde*I and *Bam*HI were then used to clone the synthesized fragment into a pET-19 vector with an N-terminal 10x His-tag. Finally, the sequence constructs were verified by Sanger sequencing services (Eton Bio) to ensure their accuracy. The sequence of the plasmid encoding the codon-optimized Pfu-Sso7d, the E602D mutant Taq polymerase, and the codon-optimized M-MLV RT have been deposited in the Zenodo open data repository [https://doi.org/10.5281/zenodo.7551378].

Detailed protocols were used to purify all proteins expressed in this work, and a description of these optimized protocols can be found in the open-access platform protocols.io [Access to each specific protocol is available in the following links: Pfu-Sso7d: https://dx.doi.org/10. 17504/protocols.io.bzusp6we, M-MLV RT: https://dx.doi.org/10.17504/protocols.io.bsernbd6, Taq DNA pol: https://dx.doi.org/10.17504/protocols.io.bya3psgn]. Briefly, the three DNA constructs were independently transformed into *E. coli* T7 expression strains BL21(DE3) (for M-MLV RT) or C41(DE3) (for Pfu-Sso7d and Taq DNA pol). Single colonies were first picked into a pre-inoculum of liquid LB media and allowed to grow overnight at 37˚C with constant agitation at 200–250 rpm. The saturated cultures were then used to inoculate 1 L of LB media and grew with constant agitation at 160–200 rpm at 37˚C until the $OD_{600}$ reached 0.8. At this point, protein overexpression was induced by adding isopropyl β-D-1-thiogalactopyranoside (IPTG) to a final concentration of 0.5 mM and incubated for 16 h at 160 rpm and 18˚C for Pfu-Sso7d and MMLV, while incubation was only 2 h at 37˚C for Taq DNA pol [60–62].

Cells were harvested by centrifugation and resuspended in a lysis buffer containing lysozyme at approximately 0.2 mg/mL. The cells were then disrupted by incubating them at room temperature for 20–30 minutes with constant agitation at 80 rpm to allow for enzymatic lysis. The resulting lysate was then sonicated to further encourage cell lysis and fragment bacterial DNA. For the thermostable enzymes Pfu-Sso7d and Taq DNA polymerase, this step was followed by a 30 min heat shock at 70–75˚C. The crude lysate was then clarified by centrifugation and the soluble fraction was collected for further purification steps [60–62].

Proteins were purified from the clarified lysate by two subsequent purification using affinity chromatography. First, a Ni-NTA IMAC purification was performed, obtaining a target protein purity >80%. The pooled fractions from IMAC were then subjected to heparin affinity chromatography to simultaneously eliminate protein and nucleic acid impurities. The purity of the eluted protein fractions was evaluated by SDS-PAGE, and the purest fractions were pooled and quantified using the Bio-Rad Protein Assay (Bio-Rad), with concentrations of the

protein solutions being estimated based on a BSA calibration curve. For long-term storage, the pooled protein fractions were adjusted to a final concentration ranging between 0.2 mg/mL and 0.6 mg/mL. They were then supplemented with 25 mM Tris-HCl pH 8.0, ~250 mM NaCl, 0.1 mM EDTA, 0.1% Nonidet P-40, and 50% glycerol. Aliquots of 200 μL were prepared and stored at -20˚C until required. Detailed storage conditions and purification protocols for each protein can be accessed on protocols.io. While our stability evaluations have confirmed enzyme functionality for up to one year, laboratories should conduct their own assessments to ensure optimal performance in their specific conditions [60–62].

## Assessment of enzymatic functionality

The functionality of DNA polymerase was evaluated by amplifying different DNA fragment sizes (122, 427, and 1067 base pairs) from a plant organism (*Arabidopsis thaliana*) unrelated to the SARS-CoV-2 virus. For these assays, 0.3 μg of homebrewed Pfu-Sso7d were compared to 0.4 U of Phusion High-Fidelity DNA Polymerase (Thermo Scientific), which was used as a positive control. Both enzymes were run in the presence of 5X Phusion HF Buffer (Thermo Scientific). A Two-Step RT-PCR reaction was then carried out to evaluate the reverse transcriptase activity of the homebrewed M-MLV-RT. For this reaction, 3 ng of home-brewed M-MLV-RT and 0.3 μg of home-brewed Pfu-Sso7d were used for cDNA synthesis and DNA amplification, respectively. A known sample of plant RNA was used in these assays, with ImProm-II RT (Promega) serving as a positive control for reverse transcriptase activity. Commercial reaction buffers ImProm-II 5X (Promega) and 5X Phusion HF Buffer (Thermo Scientific) were used, and 5 mM dithiothreitol (DTT) and 20 mM β-mercaptoethanol (BME) were tested as reducing agents.

Once the functionalities of the polymerase and reverse transcriptase enzymes were confirmed, both were evaluated in a One-Step RT-PCR using a homemade buffer composed of 150 mM Tris-HCl pH 8.4, 50 mM KCl, 50 mM NH$_4$OAc, 15 mM MgSO$_4$, 50 mM DTT, 0.5% Triton X-100, and 0.5 mg/mL BSA. This homemade buffer was optimized by individually adjusting all components within the following ranges: 100–600 mM Tris-HCl pH 7–8.5, 10–400 mM KCl, 5–250 mM NH$_4$OAc, 1–20 mM MgSO$_4$, 50–100 mM DTT, 0.1–0.5% Triton X-100, and 0.01–1 mg/mL BSA. In parallel, a One-Step RT-PCR using 0.4 μg of homemade Taq DNA polymerase instead of Pfu-Sso7d was evaluated using the same conditions described above. In both cases, 1 and 0.5 μg of plant RNA from a known sample were used as a template.

## *In vitro* transcription of N gene RNA from MERS-CoV, SARS-CoV-1 and SARS-CoV-2

Synthetic viral RNA was used for preliminary RT-PCR evaluation using homebrew M-MLV RT along with either Pfu-Sso7d or Taq DNA pol. The positive controls for the N genes from MERS-CoV, SARS-CoV-1 and SARS-CoV-2 were purchased from IDT (Cat # 10006623, 10006624, and 10006625, respectively), from which overhang PCR was performed to generate RNAP T7 transcriptional units of the full-length N genes, using the following primers: *NT_Fw* (5'-CGAAATTAATACGACTCACTATAGGGGCAACGCGATGACGATGGA TAG -3') and *T7_Nter_Rv* (5'- ACTGATCAAAAAACCCCTCAAGACCCGTTTAGAG GCCCCAAGGGGTTAT GCTAGTTAGGCCTGAGTTGAGTCAG-3').

The PCR products were purified from agarose gel electrophoresis using a Wizard Sv Gel Clean-up system (Promega) and further used in an in vitro transcription reaction at 37˚C for 16 h (Hi Scribe, NEB) using 2 μL of each ribonucleotide (100 mM), 2 μL of 10X Reaction Buffer (NEB), 0.5 μL of RNAsin (Promega), 0.5 μL of Pyrophosphatase (NEB), 2 μL of T7 polymerase Mix (NEB) and 7 μL of the N gene dsDNA linear fragment (40 ng/μL). Subsequently, a DNAse I

treatment was performed to remove the DNA template by adding 70 μL of ultrapure water, 10 μL 10X DNAse I Buffer, and 2 μL of DNAse I (NEB), followed by incubation of the sample at 37°C for 15 min. Finally, the RNA product was purified with an RNeasy kit following the provider's instructions (Qiagen), obtaining 50 μL of the viral N RNA at a concentration of 1,000 ng/μL.

## RT-qPCR analysis using synthetic RNA

Probe-based Open One-Step RT-qPCR (reaction mix and cycling conditions) and dye-based Open One-Step RT-qPCR were evaluated on *in vitro*-transcribed N gene RNA samples. Probe-based RT-qPCR was performed using specific primers for the SARS-CoV-2 nucleocapsid (N) coding gene and the human RNAse P gene (N1, N2 and RNAse P primer sets, Table 1), as well as double-quenched probes (N1, N2 and RNAse P 5' FAM / ZEN™ / 3' Iowa Black™ FQ probes). Dye-based RT-qPCR used only specific primers for the N gene and the human RNAse P gene, along with EvaGreen DNA-binding agent (Table 1). Both assays were carried out using the homemade RT-qPCR reaction buffer (150 mM Tris-HCl pH 8.4, 50 mM KCl, 50 mM $NH_4OAc$, 15 mM $MgSO_4$, 0.5% Triton X-100 and 0.5 mg/mL BSA) and 100 mM DTT. Serial dilutions in the range of $1.4 \times 10^7$ to $1.4 \times 10^1$ copies of synthetic RNA were used as templates. RT-qPCR reaction mixes were prepared just before use and kept on ice until transferred to a StepOnePlus Real-Time PCR System instrument (Applied Biosystems).

## RT-qPCR analysis using clinical samples

The Open One-Step RT-qPCR kits were validated using clinical samples that were previously labeled as positive or negative for SARS-CoV-2 in clinical or molecular virology laboratories. A total of 40 clinical samples (20 positives and 20 negatives) were reevaluated using both a commercial kit (TaqPath™ 1-Step RT-qPCR Master Mix) and our probe-based RT-qPCR reaction mix. The commercial kit was used according to the recommended conditions indicated by the supplier, while the probe-based RT-qPCR reaction mix was used following the described conditions in **Tables 2** and **3**. Both kits used primers and probes for the N1 and N2 targets of the SARS-CoV-2, as well as the human RNAse P gene target as an endogenous internal control (Table 1). Additionally, 33 samples, which had previously been labeled by clinical diagnostic services, were tested using the same commercial kit (TaqPath™ 1-Step RT-qPCR Master Mix) and our dye-based RT-qPCR reaction mix (conditions described in **S1** and S2 **Tables**).

**Table 1. Primers used in One-Step qRT-PCR reactions.**

| Name | Target | Sequence (5'→3')* |
|---|---|---|
| 2019-nCoV_N1-F | SARS-CoV-2 N gen Forward primer | GACCCCAAAATCAGCGAAAT |
| 2019-nCoV_N1-R | SARS-CoV-2 N gene Reverse primer | TCTGGTTACTGCCAGTTGAATCTG |
| 2019-nCoV_N1-P | SARS-CoV-2 N gene Probe | **FAM**–ACCCCGCAT/**ZEN**/TACGTTTGGTGGACC–**3IABkFQ**** |
| 2019-nCoV_N2-F | SARS-CoV-2 N gene Forward primer | TTACAAACATTGGCCGCAAA |
| 2019-nCoV_N2-R | SARS-CoV-2 N gene Reverse primer | GCGCGACATTCCGAAGAA |
| 2019-nCoV_N2-P | SARS-CoV-2 N gene Probe | **FAM**–ACAATTTGC/**ZEN**/CCCCAGCGCTTCAG–**3IABkF**** |
| RP-F | Human RNAse P gene Forward primer | AGATTTGGACCTGCGAGCG |
| RP-R | Human RNAse P gene Reverse primer | GAGCGGCTGTCTCCACAAGT |
| RP-P | Human RNAse P gene Probe | **FAM–**TTCTGACCT/**ZEN**/GAAGGCTCTGCGCG–**3IABkFQ**** |

* Primers and probes sequences used correspond to those described in the CDC list of Research Use Only RT-qPCR Primers and Probes [22]. Sequences may have changed according to the evolution of SARS-CoV-2.

** Probes are 5'end-labeled with the reporter molecule 6-carboxyfluorescein (**FAM**) and 3'end-labeled with a double quencher, **ZEN**™ Internal Quencher and Iowa Black® FQ (**3IABkFQ**).

**Table 2. One-Step RT-qPCR reaction mix using M-MLV RT and Taq DNA pol.**

| One-Step RT-qPCR (M-MLV/Taq) | Volume (µL) | Final Concentration |
|---|---|---|
| RNA sample | 5 | - |
| Combined Primer/Probe Mix (6,7 µM primers / 1.7 µM probe) | 1.5 | 500 nM (forward and reverse primers) 125 nM (probe) |
| 10 mM dNTPs | 0.8 | 400 nM |
| 5X Homemade Buffer | 4 | 1X |
| 100 mM DTT | 2 | 10 mM |
| Taq DNA Pol (0.4 mg/mL) | 1 | 20 ng/µL |
| M-MLV RT (0.02 mg/mL) | 1 | 1 ng/µL |
| Nuclease-Free Water | 4.7 | - |
| Total Reaction Volume | 20 | |

The One-Step RT-qPCR reaction mixes were prepared just before use and kept on ice until they were transferred to a StepOnePlus Real-Time PCR System instrument (Applied Biosystems). Cutoff points for Cq values required to determine whether a result is COVID-19 positive or negative were specified by the CDC. To report a positive result, both viral targets N1 and N2 must have a Cq value < 40. To report a negative result, both viral targets must have a Cq value $\geq$ 40. If one of the viral targets has a Cq value < 40 and the other has a Cq value $\geq$ 40, the result is reported as indeterminate.

In addition, the samples were considered valid only if the Cq value of the RNase P target was $\leq$ 35.

The specificity and sensitivity of the assays were considered in each analysis. Specificity was calculated as the number of true negatives over the sum of true negatives and false positives, whereas sensitivity was calculated as the number of true positives over the sum of true positives and false negatives.

To determine the correlation between the Cq values of positive samples from Open One-Step RT-qPCR and commercial kits, a Pearson correlation analysis was performed using the GraphPad Prism 8 software. The $r^2$ was calculated for each of these cases and significance was evaluated with an alpha = 0.05.

## Results

### Enzyme purification and standardization of One-Step RT-PCR

To develop an open-source master mix for a probe-based One-Step RT-qPCR, the following recombinant enzymes were expressed and purified: (i) a highly processive and thermostable M-MLV RT (D200N, L603W, T330P, L139P, and E607K point mutations) was used for cDNA

**Table 3. One-Step RT-qPCR cycling conditions.**

| Step | Temperature (˚C) | Duration |
|---|---|---|
| 1 | 50 | 10 min |
| 2 | 95 | 2 min |
| Repeat steps 3 and 4 by 40 cycles | | |
| 3 | 95 | 3 s |
| 4* | 55 | 30 s |

*In this step fluorescence signal acquisition occurs (FAM)

generation [59]. (ii) Taq DNA pol was used for DNA amplification [36]. (iii) To assess the performance of a dye-based RT-qPCR not dependent on probes, Pfu-Sso7d polymerase was also produced and purified [62]. Briefly, M-MLV RT, Taq DNA pol and Pfu-Sso7d were obtained by IPTG-induced protein overexpression in *Escherichia coli* cells and purified using a two-step process of nickel-nitrilotriacetic acid (Ni-NTA) immobilized metal affinity chromatography (IMAC) followed by heparin affinity chromatography (Materials and Methods section). The protein-containing fractions were pooled and analyzed by SDS-PAGE (**S1 Fig**).

The activity of the purified enzymes was tested using two experimental approaches. First, Pfu-Sso7d and M-MLV RT were used in a two-step RT-PCR assay with a known RNA sample. Different commercial reaction buffers and commonly used reducing agents, namely beta-mercaptoethanol (BME) and dithiothreitol (DTT), were tested in this experiment. The rationale behind testing two different reducing agents is the significant difference in price and stability in solution between these two chemicals [63]. A DNA band of the expected size was obtained regardless of the buffer and reducing agent used. However, the best results were obtained when DTT was added as the reducing agent (**S2A Fig**). Subsequently, a homemade 5X reaction buffer (HM-Buffer, described in Material & Methods) was tested for One-Step RT-PCR using M-MLV RT for cDNA synthesis and either Taq DNA pol (**S2B Fig**) or Pfu-Sso7d (**S2C Fig**) for DNA amplification. A band of the expected size was obtained in both One-Step RT-PCR approaches using either 0.5 or 1 µg of total DNAse-treated RNA as a template (**S2B and S2C Fig**).

## One-Step RT-qPCR validation using synthetic RNA from SARS-CoV-2

Once the components of the One-Step RT-qPCR were tested and validated, the homemade master mix was evaluated using synthetic SARS-CoV-2 RNA samples. First, the SARS-CoV-2 nucleocapsid (N) gene was *in vitro* transcribed and treated with DNAse I to remove template DNA. Subsequently, the synthetic RNA of the N gene was purified, resulting in 50 µL of RNA at a concentration of 1000 ng/µL, which was then serially diluted to obtain different target concentrations (details in Materials and Methods). Synthetic RNA of N gene from MERS-CoV and SARS-CoV-1 were also synthesized to use as negative controls. Then, a probe-based assay relying on the reverse transcriptase activity of M-MLV and the DNA polymerase activity of Taq DNA pol was performed using the *in vitro* transcribed viral RNA template. CDC-recommended primers and double-quenched probes (Center for Disease Control and Prevention, [20]) were used to amplify fragments of 72 bp (N1 pair) and 67 bp (N2 pair) within the SARS-CoV-2 N gene (details in Material & Methods section). The 5' exonuclease activity necessary for degrading the probes in this assay is provided by the Taq DNA pol enzyme. The reaction mix was prepared as indicated in **Table 2**, and the thermal cycling protocol utilized is described in **Table 3**. Representative results of the in-house probe-based One-Step RT-qPCR assay using different dilutions of synthetic RNA are shown in **Fig 1**.

As shown in **Fig 1**, the probe-based One-Step RT-qPCR assay enabled the detection of synthetic RNA copies of the SARS-CoV-2 N gen using both sets of primers (**Fig 1A and 1B**), detecting up to 10 copies of the N gene per reaction. Considering the total reaction volume of 20 µL (**Table 2**), this number of copies is equivalent to approximately 500 copies of RNA per mL, which is similar to the limit of detection (LOD) of some approved kits for SARS-CoV-2 detection [64], as well as other published open RT-qPCR methods [36].

Furthermore, no signal was obtained in the No-Template Controls (NTCs) or with synthetic RNAs from MERS-CoV and SARS-CoV-1. These results demonstrate that the locally produced enzymes and homemade master mix used in this One-Step RT-qPCR setup based on Taqman probes allow for sensitive detection of synthetic RNA from SARS-CoV-2, providing a wide dynamic range (from 10 to $10^7$ RNA copies per reaction).

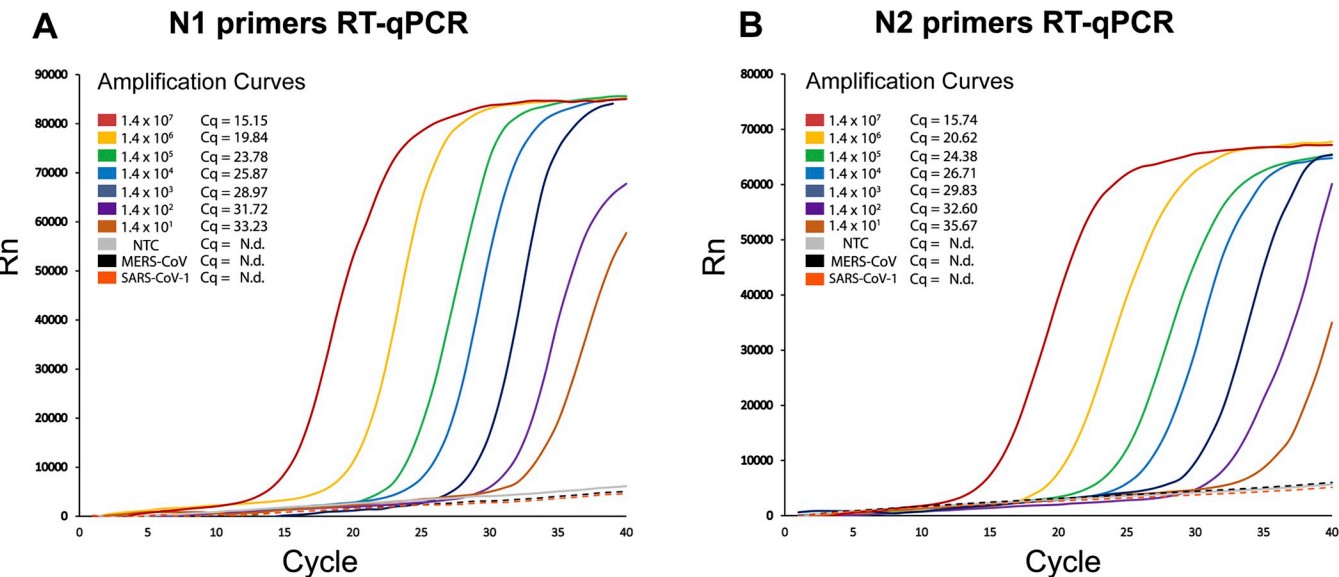

**Fig 1. CDC SARS-CoV-2 N1 and N2 probe-based One-Step RT-qPCR assays performed with synthetic RNA using homebrew M-MLV RT and Taq DNA pol.** (A-B) Representative amplification curves using N1 and N2 CDC-approved double-quenched probes. Each curve represents a specific dilution of SARS-CoV-2 synthetic N RNA used as template: $1.4 \times 10^7$ copies approximately (red line), $1.4 \times 10^6$ (yellow line), $1.4 \times 10^5$ (green line), $1.4 \times 10^4$ (light blue line), $1.4 \times 10^3$ (blue line), $1.4 \times 10^2$ (purple line), $1.4 \times 10^1$ (light brown line) and no template control (NTC, gray line). Amplification curves for synthetic N RNA from MERS-CoV (black dashed line) and SARS-CoV-1 (orange dashed line) are also indicated. Characteristic Cq values are indicated on the upper left side of each panel. N.d.: non-detected (no Cq reported).

Alternatively, a dye-based RT-qPCR using homemade M-MLV RT and Pfu-Sso7d enzymes was also assessed as a probe-independent option, avoiding the use of expensive sequence-specific fluorescently-labeled probes. This detection system, which only requires PCR primers and an intercalating dye, offers an attractive and cost-effective alternative to Taqman probes. In this assay, the EvaGreen DNA-binding agent and the same set of primers for N1 and N2 were used (Material & Methods section). The reaction buffer used for each primer set is shown in **S1 Table**, and RT-qPCR cycling conditions are described in **S2 Table**. Representative results using synthetic RNA templates are shown in **S3 Fig**. Additionally, a melting curve analysis was performed to confirm the amplification of a specific product, as the fluorescent signal generated by intercalating dye molecules (e.g., EvaGreen and SYBR Green) is not sequence-specific.

The dye-based One-Step RT-qPCR assay detected approximately 100 copies of synthetic SARS-CoV-2 RNA per reaction (equivalent to 5,000 copies of RNA per mL), which is at least one order of magnitude lower than the LOD of commercial kits [64]. However, there was non-specific amplification observed in NTC and low-concentration RNA samples, despite the efforts made to optimize the components of the RT-PCR master mix. Melting curve analysis of N1 and N2 shown in **S3 Fig** suggests that the results obtained could be due to cross-contamination with synthetic RNA in the N1 primer set and primer dimers in the N2 primer set. Taken together, these findings indicate that a probe-based One-Step RT-qPCR approach using homemade M-MLV RT and Taq DNA pol is more suitable for SARS-CoV-2 synthetic RNA detection and represents a viable alternative for further testing in the diagnosis of clinical samples.

## Validation of a Taqman probe-based One-Step RT-qPCR in clinical samples

After confirming the accuracy and sensitivity of the probe-based One-Step RT-qPCR master mix for detecting synthetic SARS-CoV-2 RNA, validation was performed using clinical

samples. CDC-recommended primers and probes that amplify the N1 and N2 targets of the SARS-CoV-2 nucleocapsid (N) protein were used, as previously described. Additionally, the human RNAse P gene target was also amplified as an endogenous internal control.

Two kits were used to evaluate 40 clinical nasopharyngeal samples: a commercial kit (Taq-Path™ 1-Step RT-qPCR Master Mix) and a homemade Taqman probe-based One-Step reaction mix using locally produced M-MLV RT and Taq DNA pol. These samples had previously been analyzed and classified in a clinical laboratory according to RT-qPCR guidelines recommended by CDC [20] (20 positives and 20 negative samples). The results obtained with the probe-based reaction mix were comparable to those of the commercial kit, as evidenced by the Cq values obtained for both kits in each of the primer/probe sets (N1, N2, and RNAse P) presented in S3 Table. None of the previously reported negative samples exhibited signal or amplification, indicating that this probe-based assay has 100% specificity or "true negative rate" (20/20 successfully assigned negative samples).

All samples that had previously been reported as positive were found to be positive in the current assay, except for one sample that could not be accurately identified as positive using either the commercial or the homemade kit. This indicates that the sensitivity of the probe-based test is 95% (19/20 positive samples were correctly identified). Additionally, all samples showed amplification for RNAse P primers/probe mix, whether using the commercial kit (mean Cq value of 27.82) or the homemade probe-based kit (mean Cq value of 27.99). Strong correlation ($r^2 = 0.9803$ and $r^2 = 0.9754$, respectively, both with a $P$-value<0.0001) was observed between the Cq values of the positive samples in both kits, regardless of whether the N1 or N2 primer/probe set was employed (Fig 2). Fig 2 also shows that all the negative samples are clustered in the section of samples where neither kit was able to detect SARS-CoV-2 N RNA (N.d: non-detected Cq value). This zone also contains sample 20 (upper right corner of Fig 2A and 2B), the positive sample that was mislabeled as negative by both kits.

Despite the non-specific amplification obtained with the dye-based One-Step RT-qPCR reaction mix (M-MLV RT/Pfu-Sso7d), 33 clinical nasopharyngeal samples were evaluated

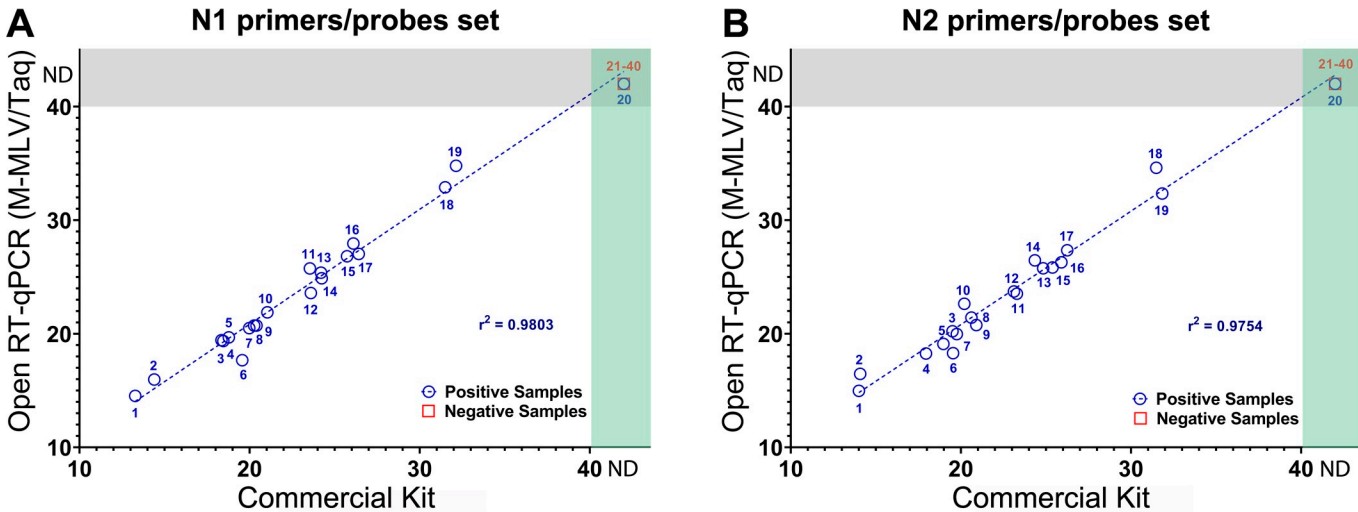

**Fig 2. Probe-based Open One-Step RT-qPCR reaction mix (M-MLV RT and Taq DNA pol) provides comparable results to commercial kits in SARS-CoV-2 clinical samples.** Scatterplot of the Cq values of positive (blue circles) and negative (red squares) samples obtained by the commercial and open RT-qPCR reaction master mixes using the N1 and N2 primer/probe sets (**A** and **B**, respectively). The numbers displayed in each sample match those displayed in Table 3 (# Sample). If a Cq value was not detected in the sample, it appears in the ND area of the graph depending on whether this occurred in the commercial kit (green rectangle), the open probe-based kit (gray rectangle), or both (intersections between the rectangles). For each combination of primers and probes, the linear trend of the positive samples is shown (blue dotted line) along with the corresponding value of $r^2$. **ND:** non-detected.

with this mix in order to assess a correlation with the TaqPath™ commercial kit. **S4 Table** displays the Cq values obtained with each of the primer sets (N1, N2, and RNAse P). The results of the study show that both reaction mixes (homemade and commercial) were able to detect the presence of SARS-CoV-2 virus in the positive samples analyzed. However, the homemade dye-based assay showed non-specific amplification, as it identified three negative samples as positive and six as indeterminate, indicating insufficient specificity for clinical analysis. In contrast, the commercial kit accurately reported all negative samples. These results demonstrate the high sensitivity of the dye-based assay, with all 24 positive samples correctly identified. However, the assay's specificity is insufficient for clinical analysis as none of the 9 negative samples were correctly identified. Moreover, the dye-based kit and the commercial kit provided comparable results for the RNAse P control, with mean Cq values of 28.04 and 28.2, respectively. Despite the inability of the dye-based RT-qPCR method to accurately report the negative samples, there was a high correlation between the Cq values of positive nasopharyngeal samples of both kits (**S4A and S4B Fig**). Therefore, once modified and optimized to increase discrimination between positive and negative samples, this dye-based test could serve as a valuable resource.

In summary, these findings demonstrate the effectiveness of the homemade, probe-based One-Step RT-qPCR reaction mix in detecting SARS-CoV-2 RNA in clinical samples and different viral synthetic RNA dilutions. Additionally, this reaction mix provides Cq values comparable to those of the TaqPath reaction mix, allowing for efficient classification of positive and negative samples. These results emphasize the potential for local production of homemade M-MLV RT and Taq DNA pol, as well as master mixes, for clinical use. This is particularly crucial for low- and middle-income countries to respond to health crises with greater autonomy and at lower economic costs.

## Discussion

Clinical laboratory testing played a crucial role in the global response to COVID-19, enabling rapid diagnosis, early intervention, and accurate disease detection and control [65]. Furthermore, open science communities that emphasize open access, open data, and open source have been essential in promoting democratic and rapid responses to the epidemic's global crisis. Given that One-Step RT-qPCR remains the widely accepted gold standard approach for detecting SARS-CoV-2 [20, 21], an open One-Step RT-qPCR method was developed to identify SARS-CoV-2 RNA in clinical samples. This open RT-qPCR approach uses recombinant enzymes and a non-proprietary buffer, which can be produced locally. While initially developed in response to the supply chain bottlenecks experienced during the peak of the COVID-19 pandemic [66], its relevance remains due to its potential cost-effectiveness and adaptability for local production, regardless of global supply chain conditions.

We used a highly processive and thermostable MMLV-RT enzyme [59] for cDNA synthesis, and two DNA polymerases for DNA amplification to perform both probe-based (Taq DNA pol [36]) and dye-based assays (Pfu-Sso7d [62]). The molecular assay, utilizing our homemade M-MLV RT and Taq DNA pol, yielded results consistent with those obtained using the Taq-Path™ commercial kit (Thermo Fisher Scientific), a commercial kit extensively utilized in clinical laboratories across Chile and globally. Our combination of locally produced enzymes demonstrated a high specificity (100%) and sensitivity (95%), accurately distinguishing between negative and positive SARS-CoV-2 cases in nasopharyngeal swab samples. Interestingly, one sample that had been previously labeled by clinical diagnostic services showed a discrepancy in both the commercial and our homemade kit. Given that these tests were conducted in parallel using both kits, potential sample degradation during transport or the

effects of freeze-thaw cycles could account for such discrepancies. Previous studies have also indicated these factors as potential sources of variation in Cq values of clinical samples [36]. Although our comparison was based on a single commercial kit, it was chosen due to its routine use and reliability in clinical diagnostics. In fact, TaqPath™ commercial kit was one of the first kits to be recommended by the CDC and approved by the FDA for SARS-CoV-2 detection [21]. While our study was limited to 40 clinical samples and a comparison with one commercial method, future work should involve multiple commercial kits and a larger sample size for enhanced validation.

On the other hand, although dye-based detection using intercalating agents such as Eva-Green is a more cost-effective alternative to probe-based assays, the probe-independent assay developed in this study using M-MLV RT and Pfu-Sso7d was not able to replicate the results of the commercial kit (**S4 Table** and **S4 Fig**). This discrepancy becomes evident when comparing the detection thresholds of our assays targeting the N gene using synthetic RNA (**Fig 1 and S3 Fig**). Specifically, the dye-based assay can detect up to 100 copies of synthetic RNA, while the probe-based assay, with its enhanced sensitivity, identifies as few as 10 copies, aligning with the detection limits of commercial kits commonly used in clinical settings [36].

In addition, when EvaGreen was used as an intercalating agent, melting curves showed non-specific amplification in NTC and low-concentration RNA samples (**S3 Fig**). This likely arises from intercalating agents binding to any dsDNA generated during the reaction, as previously described for dye-based assays using this set of primers [67, 68]. As a result, this assay had limited specificity when compared to a commercial kit (**S4 Table**), despite its high correlation with positive samples (**S4 Fig**). A potential solution for mislabeling negative samples using the dye-based approach could be to establish a cut-off cycle at which non-specific products arise, reducing the incidence of false positives without substantially affecting the LOD. Other strategies, such as using an AmpErase Uracil N-Glycosylase (UNG) or Uracil-DNA Glycosylase (UDG) step prior to PCR to control carry-over contamination, have been suggested [69, 70]. Furthermore, recent studies have emphasized the potential of melting curve analysis in molecular diagnostics. Unique melting temperatures, influenced by factors such as GC content or viral mutations, can help differentiate between various amplicons, thereby enhancing the specificity of SYBR Green assays. However, compared to probe-based approaches, these solutions require significantly more technical expertise and time, making them difficult and expensive to implement in clinical laboratories of LMICs.

Another important reason to prefer the probe-based approach is its ability to amplify multiple amplicons simultaneously, allowing for multiplex reactions. Although the use of sequence-specific fluorescently labeled probes can increase diagnostic test costs, probe-based qPCR procedures provide specificity and precision, resulting in long-term cost savings. The CDC-recommended Influenza SARS-CoV-2 Multiplex Assay, for instance, is a laboratory test that can diagnose and distinguish between influenza A, influenza B, and SARS-CoV-2 in upper or lower respiratory samples [71].

This enables laboratories to conserve critical testing resources, process more tests in a given time frame, and perform continuous flu monitoring while simultaneously screening for SARS-CoV-2. Additionally, genotyping of SARS-CoV-2 genome samples has revealed that almost all of the current COVID-19 diagnostic targets have undergone mutations [72], underscoring the need for genomic surveillance and rapid variant recognition to comprehend local epidemiology. Multiplexed reactions provide an excellent opportunity to monitor emerging SARS-CoV-2 VOC using specific probes for different genes [73–75] or different variants of the same target [76–78].

To facilitate the local production of the enzymes used in the present study, detailed purification protocols for each of them have been provided on the open-access platform protocols.io

(Materials and Methods). Protocols.io was selected for its mutual benefits. The platform promotes collaboration and recordkeeping, while also enabling users to interact with the protocol, making refinements and optimizations based on their unique insights and modifications. While striving for optimal enzyme purity, it's important to understand that the provided purification protocols were developed to balance efficacy with accessibility and simplicity, particularly for laboratories in resource-limited settings. Although the enzymes may not match the purity standards of commercially available counterparts, their functional efficacy remains robust, as demonstrated in our comparisons with commercial kits. Notably, recent articles, such as the study by Bhadra et al. (2021), have demonstrated the effective use of cellular reagents, including Taq polymerase and MMLV-RT, for RT-qPCR, even when these reagents contain other protein contaminants. The purification protocols provided in this platform were designed with the potential limitations in infrastructure and equipment availability of laboratories in LMIC in mind. To do this, the use of liquid chromatography instruments was avoided and the use of costly compounds, such as reducing agents and detergents, was minimized. The non-proprietary buffer used in the probe-based method was elaborated with cost-cutting in mind and was made up of reagents that are typically found in molecular biology and biochemistry laboratories at most academic institutions (Tris-HCL, KCl, NH4OAc, MgSO4, DTT, Triton X-100 and BSA). On the other hand, DTT emerged as a critical component in our buffer, especially for ensuring the optimal activity of both enzymes in a One-Step reaction. To ensure consistent results, DTT is supplied during the storage of M-MLV. We determined that a concentration of 50 mM DTT was optimal for our assays. Concentrations lower than this threshold were suboptimal for enzyme performance, while higher concentrations did not yield significant improvements in assay efficiency. Given our objective to reduce costs, we opted to use the minimum effective concentration. Additionally, the ReClone collaborative network simplifies the distribution of the plasmids necessary for the expression of the recombinant proteins utilized in this work, ensuring and facilitating access to these genetic resources.

While our findings are promising, it is important to acknowledge the limitations of this study. Several measures have been taken to ensure reproducibility, from cost considerations to the availability of reagents. Challenges may still be posed by some elements, such as probes, in different contexts. Our probe-free kit, developed to mitigate these costs, requires further standardization, which was not fully achieved in this study. The limited sample size, due to circumstances, is acknowledged, yet the results are believed to offer valuable insights. However, we remain committed to overcoming these challenges through networks like ReClone, fostering global laboratory connections to enhance reproducibility and validation.

The impact of SARS-CoV2 can be lessened globally in low resource settings and commercial kit shortages by using open solutions to enzyme production and distribution. In this context, our findings support an Open One-Step RT-qPCR setup compatible with Taqman probes used in clinical diagnostic laboratories. Furthermore, the results here show sensitivity and discrimination capacity comparable to commercial kits. This work represents a proof-of-principle of how open collaborative efforts can be key to respond to emergencies such as that imposed by the COVID-19 pandemic.

## Supporting information

**S1 Fig. Purification of M-MLV RT, Taq DNA pol and Pfu-Sso7d enzymes.** SDS-PAGE of M-MLV RT (A) Taq DNA pol (B) and Pfu-Sso7d (C) purification. Samples were prepared by volume; 3 μL for pellet (P), clarified lysate (CL), and flowthrough (F); 12 μL for wash (W) and elution samples from the Ni-NTA and heparin purifications. The observed molecular weight of the purified proteins matches the one predicted based on their amino acid sequences: 80

kDa for M-MLV RT, 96 kDa for Taq DNA pol, and 100 kDa for Pfu-Sso7d. The His lane in the heparin purification panel in (B) and (C) corresponds to a pooled Ni-NTA purified protein sample before heparin purification. SDS-PAGE gels in (A) and (C) were made at 10% poly-acrylamide (PA), whereas 12% PA was used for (B).
(TIF)

**S2 Fig. Homebrew M-MLV RT, Taq DNA pol and Pfu-Sso7d activity testing.** (A) Determination of M-MLV RT functionality in a conventional RT-PCR. A Two-Step RT-PCR was carried out using M-MLV RT for cDNA synthesis and Pfu-Sso7d for DNA amplification. ImProm-II 5X Reaction Buffer (Promega) and 5X Phusion HF Buffer (Thermo Scientific) were used. Additionally, dithiothreitol (DTT) and β-mercaptoethanol (BME) were tested as reducing agents. A positive control using ImProm II-RT in the 5X Reaction Buffer (Promega) was added. (B) One-Step RT-PCR using M-MLV RT for cDNA synthesis and Taq DNA pol for DNA amplification in a homemade 5X reaction buffer (composition described in Material & Methods) reaction buffer. Different amounts of total RNA treated with DNAse (1 and 0.5 μg) were used as a template. (C) One-Step RT-PCR using M-MLV RT for cDNA synthesis and Pfu-Sso7d for DNA amplification in the same conditions described in (B). The expected band size (~853 bp fragment), was obtained in all the samples. NTC: No template control. -RT: No RT control. M: DNA ladder 1 Kb (NEB).
(TIF)

**S3 Fig. CDC SARS-CoV-2 N1 and N2 One-Step RT-qPCR assays performed with synthetic RNA using homebrew M-MLV RT and Pfu-Sso7d.** (A-B) Representative amplification curves using EvaGreen as DNA intercalating dye. Each curve represents a specific dilution of SARS-CoV-2 synthetic N RNA used as template: 1.4 x 107 copies approximately (red line), 1.4 x 106 (yellow line), 1.4 x 105 (green line), 1.4 x 104 (light blue line), 1.4 x 103 (blue line), 1.4 x 102 (purple line) and no template control (NTC, gray line). Characteristic Cq values are indicated on the upper left side of each panel in order to evaluate if amplification curves correspond to single amplicons (peaks); N1 and N2 melting curve analyses are shown in panels C and D, respectively.
(TIF)

**S4 Fig. Negative samples cannot be successfully identified using the dye-based Open One-Step RT-qPCR reaction mix (M-MLV RT and Pfu-Sso7d).** Scatterplot of the Cq values of positive (blue circles) and negative (red squares) samples obtained by the commercial and open RT-qPCR reaction master mixes using the N1 and N2 primer sets (A and B, respectively). The numbers displayed in each sample match those displayed in S3 Table (# Sample). If a Cq value was not detected in the sample, it appears in the ND area of the graph depending on whether this occurred in the commercial kit (green rectangle), the open dye-based kit (gray rectangle), or both (intersections between the rectangles). For each combination of primers, the linear trend of the positive samples is shown (blue dotted line) along with the corresponding value of r2. ND: non-detected.
(TIF)

**S1 Table. Dye-based RT-qPCR reaction mix using M-MLV RT and Pfu-Sso7d.**
(DOCX)

**S2 Table. Dye-based One-Step RT-qPCR cycling conditions.**
(DOCX)

**S3 Table. Comparative Cq data between a commercial RT-qPCR kit and an Open RT-qPCR method based on homebrew M-MLV RT and Taq DNA pol.** Comparative Cq data for

the TaqPath One-Step RT-qPCR kit and the probe-based open RT-qPCR reaction mix. Assigned sample number (# Sample) and clinical sample identifier (ID) are displayed. The clinical reports of the samples before they were re-tested by the two kits are also indicated in parentheses. The sample whose report was altered is denoted in bold. (-): negative samples, (+): positive samples, ND: non-detected.
(DOCX)

**S4 Table. Comparative Cq data between a commercial RT-qPCR kit and an Open RT-qPCR method based on homebrew M-MLV RT and Pfu-Sso7d.** Comparative Cq data for the TaqPath One-Step RT-qPCR kit and the dye-based open RT-qPCR reaction mix. Assigned sample number (# Sample) and clinical sample identifier (ID) are displayed. The clinical reports of the samples before they were re-tested by the two kits are also indicated in parentheses. The samples whose reports were altered are denoted in bold. (-): negative samples, (+): positive samples, ND: non-detected.
(DOCX)

**S1 Raw images.**
(PDF)

## Acknowledgments

Thanks to Gina Dailey, Thomas Graham, Robert Tjian and Xavier Darzacq (University of California Berkeley) for sharing the Taq DNA pol plasmid. We also acknowledge the gracious help of Amparo Nuñez during the protein purification processes in the final stages of this work. We thank members of the gLAMP consortium, ReClone forum (https://reclone.org/), Open Bioeconomy Lab (https://openbioeconomy.org/) and the JOGL OpenCOVID community (https://app.jogl.io/program/opencovid19) for the advice and for sharing relevant information for the establishment of diagnostic reactions for SARS-CoV-2. Finally, we also thank Laura Delgado and Alejandro Fonseca for helping with the manuscript.

## Author Contributions

**Conceptualization:** Ariel Cerda, César A. Ramírez-Sarmiento, Fernán Federici.

**Formal analysis:** Ariel Cerda, Maira Rivera, Grace Armijo, Alexander J. Brown, Marcelo Cortez-San Martín, César A. Ramírez-Sarmiento.

**Funding acquisition:** César A. Ramírez-Sarmiento, Fernán Federici, Rodrigo A. Gutiérrez.

**Investigation:** Ariel Cerda, Maira Rivera, Grace Armijo, Catalina Ibarra-Henriquez, Javiera Reyes, Paula Blázquez-Sánchez, Javiera Avilés, Aníbal Arce, Aldo Seguel, Yesseny Vásquez, Marcelo Cortez-San Martín, Francisco A. Cubillos, Patricia García, Marcela Ferres, César A. Ramírez-Sarmiento.

**Methodology:** Ariel Cerda, Maira Rivera, Catalina Ibarra-Henriquez, Javiera Reyes, Paula Blázquez-Sánchez, Javiera Avilés, Aníbal Arce, Aldo Seguel, Yesseny Vásquez, Marcelo Cortez-San Martín, Francisco A. Cubillos, Patricia García, Marcela Ferres.

**Project administration:** Fernán Federici, Rodrigo A. Gutiérrez.

**Resources:** César A. Ramírez-Sarmiento, Fernán Federici, Rodrigo A. Gutiérrez.

**Supervision:** César A. Ramírez-Sarmiento, Fernán Federici, Rodrigo A. Gutiérrez.

**Validation:** Ariel Cerda, Maira Rivera.

**Writing – original draft:** Ariel Cerda, Grace Armijo.

**Writing – review & editing:** Ariel Cerda, Maira Rivera, Grace Armijo, Alexander J. Brown, Francisco A. Cubillos, César A. Ramírez-Sarmiento, Fernán Federici, Rodrigo A. Gutiérrez.

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
