## [Decision Letter · Decision Letter 0]

6 Sep 2023

PONE-D-23-16856An Open One-Step RT-qPCR for SARS-CoV-2 detectionPLOS ONE

Dear Dr. Gutierrez, 

Thank you for submitting your manuscript to PLOS ONE. After careful consideration, we feel that it has merit but does not fully meet PLOS ONE’s publication criteria as it currently stands. Therefore, we invite you to submit a revised version of the manuscript that addresses the points raised during the review process.

We look forward to receiving your revised manuscript.

Kind regards,

Haitham Mohamed Amer, PhD

Academic Editor

PLOS ONE

Additional Editor Comments:

Reviewers' comments:

Reviewer's Responses to Questions

**Comments to the Author**

1. Is the manuscript technically sound, and do the data support the conclusions?

Reviewer #1: Yes

Reviewer #2: Yes

2. Has the statistical analysis been performed appropriately and rigorously? 

Reviewer #1: N/A

Reviewer #2: Yes

3. Have the authors made all data underlying the findings in their manuscript fully available?

Reviewer #1: Yes

Reviewer #2: Yes

4. Is the manuscript presented in an intelligible fashion and written in standard English?

Reviewer #1: Yes

Reviewer #2: Yes

5. Review Comments to the Author

Reviewer #1: The authors propose an RT-qPCR method for SARS-CoV-2 RNA detection that is free of commercial and patented systems for use in countries with limited resources. I think this study is original and worthwhile. However, the authors should specify in the abstract that only 40 clinical samples (20 of which were negative) were tested and the results were compared with a single commercial method (in the abstract we expect several methods because the word kits is plural). This seems to me to be a major limitation of this study, even if the initial idea is laudable. It needs to be better discussed.

Table 3 should be prsented as supplementary table and a summary should be provided in the main text.

The epidemiological data (October 2022) proposed in the introduction should be updated.

Reviewer #2: The effort to generate open access resources to detect SARS-CoV-2 RNA is plausible. Homemade recombinant proteins provide a low-cost alternative to produce diagnostic kits for viral infections and diseases. This strategy is really helpful in overcoming oversaturation and scarcity of testing supplies in developing countries.

Different commercial testing kits are available at a wide range of prices. Many of them are becoming cheaper than months ago. Open access kits must be as accurate as the commercial ones. Nowadays, some open protocols for testing are available.

In this work, the home-produced enzymes M-MLV RT, Pfu-Sso7d, and Taq DNA pol, are not pure as expected for qPCR assays. Supplemental Fig1 shows a possible enzyme degradation or the presence of many contaminants in the sample. High quality proteins should be produced to maintain confident and accurately protein activity. The purification protocol must be optimized to improve the protein purity/quality. Confident long-term storage is mandatory. What is the method for protein storage? 50% of glycerol at -20C or small snap freezing aliquots at -80C? Did the authors carry out protein expression and purification optimization? Temperature, Time, and IPTG concentration should be optimized to assure optimal conditions.

Even though protein function was tested (S Fig2), the poor enzyme purity could affect reproducibility and accurate results. Protein stability is also a concern in short, mid, and long-term storage. The main goal of this study is to provide a new tool for SARS-CoV-2 detection, so the recombinant enzyme production/quality plays a key role. Did the authors follow the protein stability in time? How?

The authors explain that their open RT-qPCR master mix was evaluated whit no significant differences (Based on Cq values) compared with commercial RT-qPCR kit. In line 339, the authors suggest that samples were retested, but they are unclear enough about the details. What was the n for retesting assays? What is the discrepancy that they found?

A nonspecific amplification was observed in NTC (No Template Control) despite the optimization attempts when they used a dye-based assay. It also produced false positives or indeterminate when negative samples were analyzed. The authors mentioned this tool could be a valuable resource after optimization (line 309), but they do not show strategies or preliminary data to affirm this conclusion. They talked about the higher detection limit than the probe-based assay, but dye-based assay was not able to replicate the results of the commercial kit and it showed a non-specific amplification.

The probe-based assay is valuable due to its ability to amplify multiple amplicons simultaneously.

The homemade buffer is based Tris-HCl. How did the authors evaluate the different pH values to determine the optimum at pH 8.4? Showing these results could be valuable as a control for reproducibility in other laboratories. Tris buffers are highly sensitive for temperature/pH variations. How did the authors determine the best DTT concentration? What is the difference at lower concentrations?

6. PLOS authors have the option to publish the peer review history of their article (what does this mean?). If published, this will include your full peer review and any attached files.

Reviewer #1: **Yes: **Sylvie Pillet

Reviewer #2: No

---

## [Author Response · Author response to Decision Letter 0]

20 Oct 2023

Reviewer #1:

1.The authors propose an RT-qPCR method for SARS-CoV-2 RNA detection that is free of commercial and patented systems for use in countries with limited resources. I think this study is original and worthwhile. 

Thank you for recognizing the originality and value of our study. We aimed to develop an RT-qPCR method for SARS-CoV-2 RNA detection that can be a viable alternative for countries with limited resources. Your positive feedback reinforces our belief in the potential impact of our work.

2.However, the authors should specify in the abstract that only 40 clinical samples (20 of which were negative) were tested and the results were compared with a single commercial method (in the abstract we expect several methods because the word kits is plural). This seems to me to be a major limitation of this study, even if the initial idea is laudable. It needs to be better discussed.

Thank you for highlighting the clarity needed in our abstract and the study's limitations. We adjusted the abstract to specify the number of clinical samples tested and the single commercial method used for comparison. The emphasis has been shifted to highlight the diagnostic application of our proposed method in a real-world clinical context rather than the comparison with multiple commercial kits. We acknowledge the potential limitations of comparing our method with only one commercial kit. The choice was made based on our region's widespread use and the recognized reliability of that particular commercial method. However, we understand the implications of this choice and have addressed this limitation in the discussion section of the manuscript. We have emphasized the potential for further validation against other commercial kits to provide a more comprehensive evaluation in future studies.

3.Table 3 should be presented as a supplementary table and a summary should be provided in the main text.

Thank you for the recommendation. We have now relocated Table 3 to the Supplementary Materials, designating it as Supplementary Table 3. Furthermore, we ensured that the Results section concisely describes the pertinent information initially shown in this table. The summarized data from Supplementary Table 3 is also visually represented in Figure 2, providing a clear and quick reference for readers within the article's main content.

4.The epidemiological data (October 2022) proposed in the introduction should be updated.

Thank you for pointing out the outdated epidemiological data. We have now updated the introduction with the most recent information from the WHO report dated 1 September 2023. The reference can be found here: https://www.who.int/publications/m/item/weekly-epidemiological-update-on-covid-19---1-september-2023.

Reviewer #2: 

1.The effort to generate open access resources to detect SARS-CoV-2 RNA is plausible. Homemade recombinant proteins provide a low-cost alternative to produce diagnostic kits for viral infections and diseases. This strategy is really helpful in overcoming oversaturation and scarcity of testing supplies in developing countries.

We thank you for your positive remarks and for recognizing the significance of open-access resources in diagnostic efforts, not just for SARS-CoV-2, but for broader applications. Our commitment is rooted in the vision of providing cost-effective and efficient solutions, especially in contexts where supply shortages and oversaturation can be challenging. Your acknowledgment emphasizes the importance and relevance of our research in this domain.

2.Different commercial testing kits are available at a wide range of prices. Many of them are becoming cheaper than months ago. Open access kits must be as accurate as the commercial ones. Nowadays, some open protocols for testing are available.

We appreciate your observation of the evolving landscape of diagnostic kits. While commercial kits are becoming more affordable and diverse, our emphasis extends beyond immediate cost considerations. As the urgency of the pandemic subsides, the production of these commercial kits may decrease, impacting their global availability. Our open approach ensures sustained accessibility and adaptability, even changing market dynamics.

Moreover, the versatility of our kit allows for potential standardization to detect other RNA viruses, making it a valuable tool beyond the current pandemic. We have provided detailed purification protocols on the open-access platform protocols.io, designed with the infrastructure constraints of low and middle-income countries in mind. The non-proprietary buffer and the strategic selection of reagents underscore our commitment to accessibility and affordability. Collaborative platforms like ReClone further democratize access to essential genetic resources. Our initiative complements commercial offerings to fortify global diagnostic resilience and foster collaborative innovation.

3.In this work, the home-produced enzymes M-MLV RT, Pfu-Sso7d, and Taq DNA pol, are not pure as expected for qPCR assays. Supplemental Fig1 shows a possible enzyme degradation or the presence of many contaminants in the sample. High quality proteins should be produced to maintain confident and accurately protein activity. The purification protocol must be optimized to improve the protein purity/quality. 

Thank you for your insightful observation regarding the apparent enzyme purity in Supplemental Figure 1. In this regard, the production of the enzymes using simplified purification protocols, which indeed can include traces of protein contaminants, does not seek to replicate the purity and homogeneity of enzymatic solutions commercially available but to provide fast and affordable solutions, especially tailored for low-resource settings. Such low-income settings lack instrumentation for producing high-quality protein solutions, such as HPLCs and FPLCs, and expensive size exclusion chromatography columns that would enable quality control of the purified enzymes.

Moreover, recent articles published in this journal have demonstrated that the use of cellular reagents, namely lyophilized bacteria overexpressing proteins of interest, which are full of other protein contaminants in the cellular extracted, performed comparably to commercial enzymes for RT-qPCR (Bhadra et al., 2021, https://doi.org/10.1371/journal.pone.0252507). Notably, the enzymes tested in these preparations include those central to our study: Taq polymerase and MMLV-RT. In our work, functional validations have indicated that our enzymes, despite the visualized impurities, retain adequate activity for our RT-qPCR assays. Our purified enzymes offer similar detection limits, sensitivity, and accuracy compared to an equivalent commercial kit routinely employed in diagnostics.

Thank you for highlighting this aspect. We value your feedback and will incorporate this discussion and justification into our manuscript to delineate the focus and rationale behind our chosen methods.

4.Confident long-term storage is mandatory. What is the method for protein storage? 50% of glycerol at -20C or small snap freezing aliquots at -80C? Did the authors carry out protein expression and purification optimization? Temperature, Time, and IPTG concentration should be optimized to assure optimal conditions.

We appreciate your insightful questions regarding our protein storage and expression conditions. Our proteins are stored in a solution containing 25 mM Tris-HCl pH 8.0, ~250 mM NaCl, 0.1 mM EDTA, 0.1% Nonidet P-40, and 50% glycerol. We prepare 200 µL aliquots of the enzyme at concentrations ranging between 0.2 mg/mL and 0.6 mg/mL and store them at -20 °C until use.

As for the optimal conditions for protein overexpression, we want to clarify that the temperature, time, bacterial strains, and IPTG concentration employed in our study result from thorough optimization. While these details are extensively described on the protocols.io platform, we have also included them in the Materials and Methods section of our manuscript for the convenience of our readers.

5.Even though protein function was tested (S Fig2), the poor enzyme purity could affect reproducibility and accurate results. Protein stability is also a concern in short, mid, and long-term storage. The main goal of this study is to provide a new tool for SARS-CoV-2 detection, so the recombinant enzyme production/quality plays a key role. Did the authors follow the protein stability in time? How?

Thank you for raising this valid concern. We acknowledge that enzyme purity is a significant factor in many biochemical applications. However, as previously discussed, our primary aim was to achieve functional efficacy rather than utmost purity. Despite potential contaminants, the enzymes have shown consistent performance in our assays.

Given the ongoing demand for these enzymes, we consistently purified large quantities, ensuring a continuous supply and minimizing the elapsed time between production and use. This consistent turnover has allowed us to conduct stability tests for up to 6 months and one year using synthetic RNA as input, with the enzymes demonstrating sustained functionality over these periods. However, we understand that the stability might vary based on the specific storage and usage conditions, especially in the diverse environments of the target countries. In light of your feedback, we have updated our accompanying documentation to specify that enzyme stability has been confirmed for one year. Nevertheless, we underscore the importance for end-users to evaluate enzyme stability within their own laboratory environments rigorously.

6.The authors explain that their open RT-qPCR master mix was evaluated with no significant differences (Based on Cq values) compared with commercial RT-qPCR kit. In line 339, the authors suggest that samples were retested, but they are unclear enough about the details. What was the n for retesting assays? What is the discrepancy that they found?

Thank you for pointing out the need for clarity on the term "retested." We apologize for any confusion arising from our use of the term. To provide a more precise context:

When we mentioned "retested," we referred to the parallel testing of clinical samples, which had already been labeled by clinical diagnostic services, using our developed kit and a commercial kit (TaqPath™ 1-Step RT-qPCR Master Mix). This parallel testing was conducted based on the recommendation of the entities that supplied the samples. Given that these samples had been stored for some time prior to our analysis, there was a potential for a shift in the Ct values from what was initially reported by the clinical laboratories.

It is worth noting that in certain instances, discrepancies were observed. For instance, with sample 20 (upper right corner of Fig 2A and 2B), previously labeled by the clinical service, our kit, and the commercial kit mislabeled it during parallel testing. Such discrepancies underscore the importance of considering factors like sample transportation and storage conditions when utilizing samples for such evaluations.

We have revised the manuscript to more clearly articulate this process and its implications to ensure further understanding.

7.A nonspecific amplification was observed in NTC (No Template Control) despite the optimization attempts when they used a dye-based assay. It also produced false positives or indeterminate when negative samples were analyzed. The authors mentioned this tool could be a valuable resource after optimization (line 309), but they do not show strategies or preliminary data to affirm this conclusion. They talked about the higher detection limit than the probe-based assay, but dye-based assay was not able to replicate the results of the commercial kit and it showed a non-specific amplification. The probe-based assay is valuable due to its ability to amplify multiple amplicons simultaneously.

We appreciate your observations regarding the challenges associated with the dye-based RT-qPCR assay. The non-specific amplification, particularly in the No Template Control (NTC), has been a recognized challenge. While this phenomenon could be attributed to the nature of intercalating agents, it is a concern we share. 

Despite these challenges, a significant aspect of our findings lies in the strong correlation between the Cq values of positive nasopharyngeal samples obtained with our dye-based assay and those from a commercial kit. This correlation highlights the potential of our assay, given further optimization. In essence, these results can serve as preliminary evidence, underscoring the assay's potential, which is why we presented them in our manuscript, even if they were not the primary focus. To address the concerns about optimization strategies, we did discuss potential avenues in the manuscript, such as establishing a cut-off cycle to manage false positives and implementing techniques like UNG or UDG to combat contamination. These strategies could enhance the specificity of the dye-based assay, making it more applicable in diverse settings.

There has been a misunderstanding in our initial presentation regarding the detection limit. When we mention the dye-based assay, we mean that it has a detection threshold of 100 copies of synthetic RNA, whereas the probe-based assay can detect as few as 10 copies. In other words, while the numeric value for the dye-based assay's detection limit is higher, it indicates a reduced sensitivity compared to the probe-based assay. We have clarified this distinction in our revised manuscript to avoid further confusion.

Lastly, we concur with your emphasis on the probe-based approach's capacity for multiplex reactions. It is a distinct advantage that resonates with the current diagnostic needs, further justifying our decision to prioritize it in our study. Once again, your feedback has been instrumental, and we are grateful for the thorough review. We have concerted efforts to ensure clarity and accuracy in our revised manuscript.

8.The homemade buffer is based Tris-HCl. How did the authors evaluate the different pH values to determine the optimum at pH 8.4? Showing these results could be valuable as a control for reproducibility in other laboratories. Tris buffers are highly sensitive for temperature/pH variations. How did the authors determine the best DTT concentration? What is the difference at lower concentrations?

Our homemade buffer's composition was meticulously optimized. After confirming the functionality of the polymerase and reverse transcriptase enzymes, we embarked on an extensive optimization process for our buffer. We adjusted each component individually within the following ranges: 100-600 mM Tris-HCl pH 7-8.5, 10-400 mM KCl, 5-250 mM NH4OAc, 1-20 mM MgSO4, 50-100 mM DTT, 0.1-0.5% Triton X-100, and 0.01-1 mg/mL BSA. Given the emergency health context and limited resource conditions, our multidisciplinary team utilized what was readily available. We employed plant RNA from a known sample as a reference template for our optimization, using 1 and 0.5 µg quantities.

The choice of a Tris-HCl buffer at pH 8.4 was influenced by the recent work of Graham et al. (2021, https://doi.org/10.1371/journal.pone.0246647). Gina Dailey, Thomas Graham, Robert Tjian, and Xavier Darzacq (University of California, Berkeley) graciously provided us with the Taq DNA pol plasmid. Based on their recent publication, they recommended the pH conditions, which described an open-source RT-qPCR method using a similar buffer composition. Their protocol served as a valuable starting point for our research. On the other hand, DTT emerged as a critical component in our buffer, especially for ensuring the optimal activity of both enzymes in a One-Step reaction. To ensure consistent results, DTT is supplied during the storage of M-MLV. We determined that a concentration of 50 mM DTT was optimal for our assays. Concentrations lower than this threshold were suboptimal for enzyme performance, while higher concentrations yielded little improvements in assay efficiency. Given our objective to reduce costs, we used the minimum effective concentration.

We hope that these clarifications address your concerns. Additionally, it is worth mentioning that some adjustments, e.g., the use of different reducing agents in the enzyme buffers, can be found in our protein purification protocols on protocols.io. We have incorporated these and other relevant details into our manuscript to assist future researchers in enhancing method reproducibility.

---

## [Decision Letter · Decision Letter 1]

7 Nov 2023

PONE-D-23-16856R1An open one-step RT-qPCR for SARS-CoV-2 detectionPLOS ONE

Dear Dr. Gutierrez,

Thank you for submitting your manuscript to PLOS ONE. After careful consideration, we feel that it has merit but does not fully meet PLOS ONE’s publication criteria as it currently stands. Therefore, we invite you to submit a revised version of the manuscript that addresses the points raised during the review process.

We look forward to receiving your revised manuscript.

Kind regards,

Haitham Mohamed Amer, PhD

Academic Editor

PLOS ONE

Journal Requirements:

Reviewers' comments:

Reviewer's Responses to Questions

**Comments to the Author**

1. If the authors have adequately addressed your comments raised in a previous round of review and you feel that this manuscript is now acceptable for publication, you may indicate that here to bypass the “Comments to the Author” section, enter your conflict of interest statement in the “Confidential to Editor” section, and submit your "Accept" recommendation.

Reviewer #2: All comments have been addressed

Reviewer #3: All comments have been addressed

2. Is the manuscript technically sound, and do the data support the conclusions?

Reviewer #2: Yes

Reviewer #3: Yes

3. Has the statistical analysis been performed appropriately and rigorously? 

Reviewer #2: I Don't Know

Reviewer #3: Yes

4. Have the authors made all data underlying the findings in their manuscript fully available?

Reviewer #2: Yes

Reviewer #3: Yes

5. Is the manuscript presented in an intelligible fashion and written in standard English?

Reviewer #2: Yes

Reviewer #3: Yes

6. Review Comments to the Author

Reviewer #2: The authors have adequately addressed the comments raised in my previous review. They modified the manuscript, clarifying and adding pertinent information.

Reviewer #3: This a very important study, which describes the effectiveness of a homemade, probe-based One-Step RT-qPCR reaction mix in the detection of SARS-CoV-2 RNA in clinical samples. Furthermore, it proposed an economically viable production of homed-made M-MLV RT, Taq DNA pol and master mixes for clinical use.

In page 6, line 151 and 153; Page 7 165-167: Please include catalogue numbers of the kits used.

In Page 7, line 163: The statement, “were collected from symptomatic cases of COVID-19 consultants at the UC-Christus Health” …..., is an incomplete sentence. Do you mean that samples were collected from consultants who were COVID-19 positive or, or from COVID-19 patients, or consultants attending to COVID-19 patients? You may wish to rephrase the sentence for easy comprehension.

The presentation of methodology and results are commendable. However, the authors did not describe the ‘Limitation of the study’. What is/are the limitation(s) of the study? Can the findings be replicated in other independent laboratories outside of the home laboratory or home country? Please make 1-2 statements on study limitations.

Also, recommendation to include interlaboratory ring trials in some low to middle-income countries may be necessary (optional!)

7. PLOS authors have the option to publish the peer review history of their article (what does this mean?). If published, this will include your full peer review and any attached files.

Reviewer #2: No

Reviewer #3: No

---

## [Author Response · Author response to Decision Letter 1]

18 Dec 2023

Dear Dr. Amer,

 Please find enclosed the revised version of our manuscript entitled “An Open One-Step RT-qPCR for SARS-CoV-2 detection” [PONE-D-23-16856] - [EMID:486baa91abbf9866], by Ariel Cerda, Maira Rivera, Grace Armijo, Catalina Ibarra-Henriquez, Javiera Reyes, Paula Blázquez-Sánchez, Javiera Avilés, Aníbal Arce, Aldo Seguel, Alexander J. Brown, Yesseny Vásquez, Marcelo Cortez-San Martín, Francisco A. Cubillos, Patricia García, Marcela Ferres, César A. Ramírez-Sarmiento, Fernán Federici & Rodrigo A. Gutiérrez.

Thank you for the opportunity to revise our manuscript. We responded to the requirements requested by you as an editor and constructively addressed the points raised by the reviewer 3, as detailed below. 

Reviewer #3: This a very important study, which describes the effectiveness of a homemade, probe-based One-Step RT-qPCR reaction mix in the detection of SARS-CoV-2 RNA in clinical samples. Furthermore, it proposed an economically viable production of homed-made M-MLV RT, Taq DNA pol and master mixes for clinical use.

Thank you for your positive comments and recognition of our work in developing a homemade One-Step RT-qPCR mix for SARS-CoV-2 detection. We believe this approach is vital for resource-limited settings and hope it significantly contributes to the scientific and medical communities. Your feedback has been invaluable in emphasizing the impact of our study.

In page 6, line 151 and 153; Page 7 165-167: Please include catalogue numbers of the kits used.

We have addressed your suggestion and updated the manuscript accordingly. The catalogue numbers for the kits have now been included in the specified sections (page 6, lines 151 and 153; page 7, lines 165-167). 

In Page 7, line 163: The statement, “were collected from symptomatic cases of COVID-19 consultants at the UC-Christus Health” …..., is an incomplete sentence. Do you mean that samples were collected from consultants who were COVID-19 positive or, or from COVID-19 patients, or consultants attending to COVID-19 patients? You may wish to rephrase the sentence for easy comprehension.

We have amended the text to more precisely reflect that the samples were obtained from individuals who sought medical consultation at UC-Christus Health, displaying symptoms consistent with COVID-19. The modified sentence now reads: “On the other hand, 33 nasopharyngeal swabs in Universal Transport Medium (UTM) were collected from 17 to 22 July 2020 at the UC-Christus Health Network in Chile, from individuals presenting with symptoms suggestive of COVID-19.” Additionally, we have included the following detail to provide further clarity: ‘Of these, 24 were later confirmed as COVID-19 positive, while the remainder tested negative.’ We believe this revision accurately depicts both the context of the patient consultations and the results of their COVID-19 tests. These same collected samples were analyzed using our dye-based RT-qPCR reaction mix for the purpose of comparing with the results obtained from clinical analyses.

The presentation of methodology and results are commendable. However, the authors did not describe the ‘Limitation of the study’. What is/are the limitation(s) of the study? Can the findings be replicated in other independent laboratories outside of the home laboratory or home country? Please make 1-2 statements on study limitations.

Regarding limitations, we have added a paragraph to the discussion section acknowledging the challenges in replicating our findings in independent laboratories, particularly due to probe costs and the need for further standardization of our probe-free kit. We also acknowledge the relatively small sample size of our study. Furthermore, we highlight the role of collaborative networks like ReClone in facilitating reproducibility and global collaboration in similar assays.

Also, recommendation to include interlaboratory ring trials in some low to middle-income countries may be necessary (optional!)

We completely agree that this would greatly enhance the robustness and applicability of our findings. It aligns perfectly with our goal of establishing collaborative networks, which we are actively working on. We have added a section to our manuscript discussing this aspiration and its future potential. Due to logistical constraints at the time of this study, such trials were not feasible, but they are an important focus for our future research. 

Please do not hesitate to contact us if you have any additional comments about this manuscript.

---

## [Decision Letter · Decision Letter 2]

28 Dec 2023

An open one-step RT-qPCR for SARS-CoV-2 detection

PONE-D-23-16856R2

Dear Dr. Gutierrez,

We’re pleased to inform you that your manuscript has been judged scientifically suitable for publication and will be formally accepted for publication once it meets all outstanding technical requirements.

Kind regards,

Haitham Mohamed Amer, PhD

Academic Editor

PLOS ONE

---

## [Editor Report · Acceptance letter]

11 Jan 2024

PONE-D-23-16856R2 

PLOS ONE

Dear Dr. Gutiérrez, 

I'm pleased to inform you that your manuscript has been deemed suitable for publication in PLOS ONE. Congratulations! Your manuscript is now being handed over to our production team.

Kind regards, 

on behalf of

Dr. Haitham Mohamed Amer 

Academic Editor

PLOS ONE